# Communities as 'renewable energy' for healthcare services? a multimethods study into the form, scale and role of voluntary support for community hospitals in England

Angela Ellis Paine ,[1] Daiga Kamerāde,[2] John Mohan,[3] Deborah Davidson[4]

¹Third Sector Research Centre, School of Social Policy, University of Birmingham, Birmingham, UK
²School of Health and Society, University of Salford, Salford, UK
³Third Sector Research Center (TSRC), University of Birmingham, Birmingham, UK
⁴Health Services Management Centre, University of Birmingham, Birmingham, UK

**Correspondence to**
Dr Angela Ellis Paine;
a.ellispaine@bham.ac.uk

## ABSTRACT

**Objective**  To examine the forms, scale and role of community and voluntary support for community hospitals in England.

**Design**  A multimethods study. Quantitative analysis of Charity Commission data on levels of volunteering and voluntary income for charities supporting community hospitals. Nine qualitative case studies of community hospitals and their surrounding communities, including interviews and focus groups.

**Setting**  Community hospitals in England and their surrounding communities.

**Participants**  Charity Commission data for 245 community hospital Leagues of Friends. Interviews with staff (89), patients (60), carers (28), volunteers (35), community representatives (20), managers and commissioners (9). Focus groups with multidisciplinary teams (8 groups across nine sites, involving 43 respondents), volunteers (6 groups, 33 respondents) and community stakeholders (8 groups, 54 respondents).

**Results**  Communities support community hospitals through: human resources (average=24 volunteers a year per hospital); financial resources (median voluntary income = £15 632); practical resources through services and activities provided by voluntary and community groups; and intellectual resources (eg, consultation and coproduction). Communities provide valuable supplementary resources to the National Health Service, enhancing community hospital services, patient experience, staff morale and volunteer well-being. Such resources, however, vary in level and form from hospital to hospital and over time: voluntary income is on the decline, as is membership of League of Friends, and it can be hard to recruit regular, active volunteers.

**Conclusions**  Communities can be a significant resource for healthcare services, in ways which can enhance patient experience and service quality. Harnessing that resource, however, is not straight forward and there is a perception that it might be becoming more difficult questioning the extent to which it can be considered sustainable or 'renewable'.

## INTRODUCTION

A growing recognition of a need to find new ways to deliver public services has contributed to an increasing emphasis on involving communities and individual volunteers in the National Health Service (NHS). NHS England's Five Year Forward View, which in 2014 set out a vision for the future of the NHS, identified communities and patients as 'renewable energy'.[1] This suggests they are seen as a sustainable resource that can be naturally replenished to support the NHS. More recently, NHS England's 'NHS Long Term Plan' has committed to supporting the aim of doubling the number of volunteers in the NHS.[2] Some have gone as far as to suggest the NHS would collapse without the support of volunteers.[3] More broadly, Crisp identifies voluntary and community organisations as part of the 'informal system of care', which, if strengthened, could relieve pressure on health and social care services, contributing to their sustainability.[4] Such claims, however, are often made with limited evidence. We know little about how extensive community involvement is within the NHS, what form it takes, how it varies, or to what extent this 'energy' can be considered sustainable or 'renewable'.

**Strengths and limitations of this study**

► A robust multimethods study allowing us to understand the breadth, depth and variability of community support for community hospitals.
► Original insights and data on the levels and forms of voluntary support for community hospitals, which had previously been almost non-existent.
► Limited by the availability of secondary data for the detailed analysis of levels of volunteering (particularly outside of Leagues of Friends) and expenditure from charitable funds.
► Limited by its scope to consider the wider system within which community hospitals are embedded.

Community hospitals are a long-standing part of the healthcare landscape. In 2015, there were 296 community hospitals in England. Community hospitals are small (typically less than 30 beds), predominantly rural, traditionally general practitioner led and provide a varied mix of intermediate care services.[5] They often originated as cottage hospitals through local voluntary initiative.[6] Under NHS control, the traditions of voluntary support were maintained, particularly through the formation of Leagues of Friends—independent charities set up to raise funds and mobilise support. Historical accounts have highlighted substantial local commitment to individual community hospitals,[6–8] but we are aware of only one academic study of Hospital Leagues of Friends in the UK, published in 1960.[9]

Existing national survey evidence provides only a very general insight into voluntary action in the NHS. Naylor *et al* estimate that approximately 2.9 million people in England volunteer for health-related organisations and causes, but cannot break this down by specific settings.[10] Studies of voluntary income in healthcare have focused on relatively large organisations[11] or looked at data across authorities.[12] Galea *et al*'s study of volunteering in NHS Acute (secondary care) Trusts in England found that on average they involve 471 volunteers each.[13] The study highlighted variations between Trusts, but focused only on one form of voluntary support in one particular setting.

A small number of studies have considered the outcomes of certain forms of community engagement for hospitals and for the wider healthcare system. There is some evidence that volunteers can have a positive impact on healthcare, through improving patient experience, strengthening the relationships between services and community, improving public health and supporting integrated care.[10 14] Such evidence, however, is limited.

To investigate how communities support their community hospitals we draw on a multimethods study. The study offers original evidence on the scale, form and role of voluntary support for community hospitals, and variations over time. Given current and likely future pressures on resources within the NHS, we reflect on the extent to which voluntary support may be considered a sustainable form of 'renewable energy' offering important additional resources for hospitals, patients and communities.

## METHODS

We draw on evidence from a large multimethods study exploring the profile, characteristics, patient experience, community engagement and value of community hospitals.[5] One aim of this wider study—and of this paper—was to address the question: 'what do communities do to support their community hospitals?' More specifically, the intention is to explore the scale, forms, role and character of voluntary support to community hospitals and how these have changed over time. Developing a complete understanding of voluntary support required a mix of both quantitative (how much) and qualitative (how and why) data. A convergent design was employed, with the qualitative and quantitative elements conducted broadly in parallel and given equal status. Different researchers were involved in the two elements. The research followed good practice guidelines on mixed-methods studies.[15 16]

The quantitative element involved examining the scale of volunteering and financial support for community hospitals through compiling and analysing a dataset of records held by the Charity Commission (CC) (the charity regulator in England and Wales). The sample for this study was 245 Leagues of Friends (and other similar charities) in England that directly support community hospitals and for which financial information was available for at least 1 year between 1995 and 2014. The CC records included data on the finances and volunteer numbers for each charity.

We conducted a more detailed analysis of income sources and types of expenditure using data captured from the annual financial accounts of those charities whose income or expenditure exceeds the threshold (£25 000 per year) above which financial accounts are made available on the CC website. This gave data covering 358 sets of accounts across six financial years; the average number of charities with detailed accounts was 60 per financial year. At the time of conducting analysis, the latest detailed financial information available was for the financial year 2013. The number of accounts available varies from year to year because an individual charity may or may not exceed the £25 000 threshold, depending on fluctuations in its finances. All financial figures were adjusted for inflation using the Office for National Statistics Retail Prices annual index; therefore, all financial information in this paper is presented in 2014 prices.

We undertook descriptive analysis of variations and used a fixed-effects regression model[17] to examine how within-hospital charity changes in time (years) are linked to changes in the annual income within each individual charity over time, while eliminating unobserved heterogeneity—confounding effects from time-constant variables. The year variable examines any variation in the outcome that happen over time.

The qualitative element involved adopting a case study approach to explore the form, role, experiences and perceived outcomes of community support, reported here using the standards for reporting qualitative research (SRQR) guidelines.[18] Nine case studies were selected from a database of community hospitals in England, reflecting diversity in terms of: location, size, models of ownership and provision, and levels of voluntary income (from well below average to well above).

Each case study involved: scoping; a local reference group (staff and community members brought together to inform the study and reflect on emerging findings); semistructured interviews with: Trust managers and commissioners (9, across the nine cases), staff (89), volunteers (35), community stakeholders (20) and carers (28); discovery interviews with patients (60); focus groups with

multidisciplinary teams (8, across the 9 sites, involving 43 respondents), volunteers (6 groups, 33 respondents) and community stakeholders (8 groups, 54 respondents). We used purposive sampling, informed by our scoping visits, local reference groups and snowball techniques, with respondents selected due to their involvement in or knowledge of the community hospital. Further details can be found in the full study report.[5] Separate topic guides were developed for each group of respondents, each covering broadly similar topics (see online supplementary file). The fieldwork took place between November 2015 and February 2017.

Interviews and focus groups were recorded and transcribed verbatim before being imported into NVivo V.11 software as deidentified files. We conducted thematic analysis, guided by Braun and Clarke's six step process.[19] [20] After an initial reading of transcripts, and a subsequent open coding of a sample of transcripts by three members of the research team, a draft coding frame was developed collaboratively. An initial testing of the coding frame, led to codes being refined, reordered and grouped in to themes, before the final version was applied across the whole dataset. Processes were put in place to ensure consistency of coding across the team, including checking each other's coding practices. The qualitative research team had complementary specialist expertise in both health service management and in voluntary action. Themes were further defined, refined and validated during subsequent stages of analysis and reporting. Emerging findings were discussed with the wider research team and with diverse national and local stakeholders (see the Patient and public involvement (PPI) section). Together these different stages of analysis, throughout which we interacted deeply with the qualitative data, helped to ensure validity and reliability.

After using early findings from the initial quantitative analysis to inform the case study selection, data convergence took place throughout the analysis period, with regular whole-team meetings convened to discuss, triangulate and integrate emerging findings from across the different stands of analysis. Using quantitative and qualitative methods enabled us to compare and integrate findings from national data on the scale of community support with local, multistakeholder perspectives on the forms, role and meaning of support.

### Patient and public involvement

Key stakeholders—including members of the Community Hospitals Association, community hospital staff, patients, carers and the public—were involved in all stages of this study, including through a national Steering Group and Local Reference Groups. Emerging findings were discussed at national and local level through Local Reference Group meetings, Annual Learning Events and conferences. PPI enhanced both the quality, trustworthiness and credibility of the research and its impact.

## RESULTS

The study identified four, inter-related, types of support or 'resource' that communities provide for their hospitals.

### Human resource: giving time through volunteering in community hospitals

Analysis of CC records found that Community Hospital Leagues of Friends on average involved 24 volunteers (range 2–162): an estimated total of 5880 volunteers across the 245 community hospitals in the sample. This equates to between 1.4 and 2.5 full-time equivalent personnel. If we were to express the value of this using the national minimum wage, the volunteer labour provided by Leagues of Friends equates to a financial input of £15 600–£28 500 per community hospital, or £3.8 million to £6.9 million across England. Most case studies involved additional volunteers who were recruited directly by the hospitals, outside of the League of Friends, for which there is no nationally available data.

Our case studies found volunteers tended to be retired, female, white and middle class. Most volunteers become involved through word of mouth, limiting the potential pool of new volunteers. Examples of active volunteer recruitment campaigns were limited. Concerns were raised about the challenge of involving new people. While there were generally high levels of latent support for community hospitals, particularly demonstrable at times of threat to services, this could be difficult to convert into active, regular, volunteer engagement:

'I think that a lot of people would go up in arms if it was closed but whether they'd be willing to do anything about it to help I don't know. There seems to be a little core of people that would but I'm not sure that that spreads right out across the community' CH9, Carer

In one case study the League of Friends had ceased to operate as volunteers retired and they were unable to recruit new members. In another, however, volunteer involvement had been reinvigorated through the recruitment of a new League of Friends chair and active support from the hospital matron.

Volunteers were involved in a wide range of roles, from fundraising activities, through to delivering activity sessions for patients, supporting meal times, working on a reception desk, and providing a 'hospital to home' befriending service. There was a clear consensus that volunteers could not get involved in medical or personal care or access confidential data and that they were there to complement, not supplement, paid staff. Beyond this, the roles that volunteers undertook varied considerably.

Health and safety regulations and concerns about confidentiality were the most cited restrictions on volunteer involvement. In some, they were seen as reasons not to involve volunteers; in others as appropriate training requirements. Volunteer involvement was further limited by perceptions of key actors within the hospitals concerning: the capacity of patients to engage with

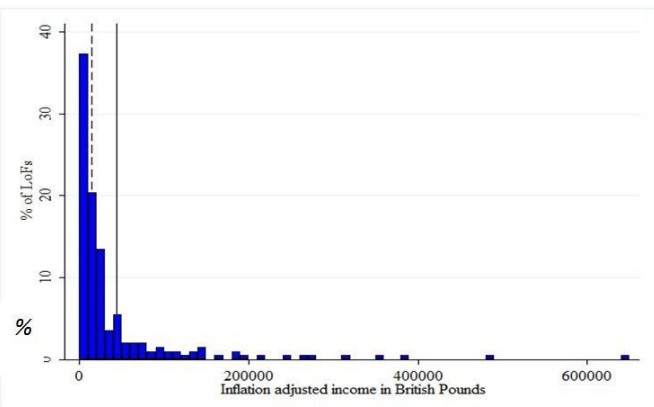

**Figure 1** LoFs income distribution in 2014. Solid line=mean (£45 387); dashed line=median (£15 632). LoFs, Leauge of Friends.

volunteers; the views of carers/families towards volunteers; capacity of staff to facilitate volunteering and the willingness of people to volunteer. Some respondents argued that such concerns were putting 'barriers up where they don't exist' and that hospitals were not always making the most of 'untapped resources' available to them.

### Financial resources: giving money through fundraising for community hospitals

Leagues of Friends provide substantial financial resource. In 2014, on average, community hospital League of Friends generated financial resources worth £45 387 (figure 1). This figure is influenced by a small number of very large outliers: the median income was £15 632. In addition, some community hospitals benefit from income generated through donations made directly to NHS Trusts: these funds are not counted in the above figures (our quantitative data is confined to League of Friends) but were reported to be significant in three case studies.

The community hospital Leagues of Friends were both relatively large and relatively stable entities within their local charitable landscapes: 40% were among the largest 100 charities in their local authorities in terms of income.

Charities generate financial resources for community hospitals through various means. Figure 2 presents an overview of income sources, derived from charity accounts from Leagues of Friends with an income or expenditure

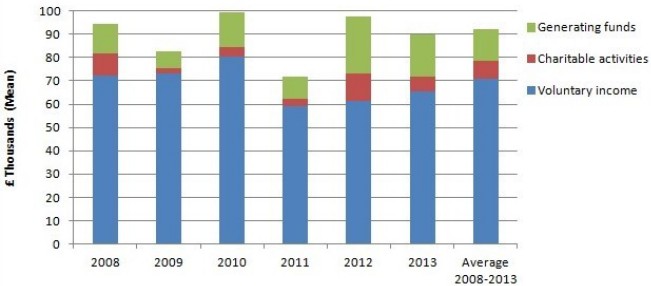

**Figure 2** Community hospital Leagues of friends' income sources, by year.

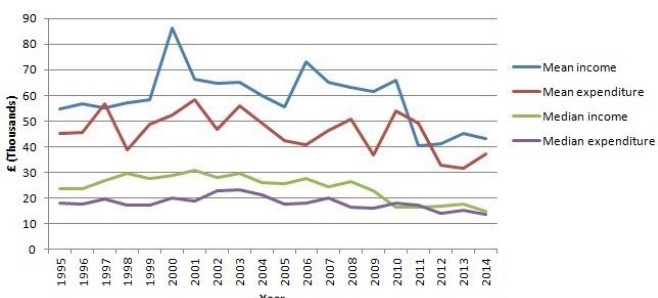

**Figure 3** Leagues of friends income and expenditure over time.

greater than £25 000 in any given year. Over a 5-year period from 2008 to 2013, nearly three-quarters (70%) of Leagues of Friends' income was 'voluntary income', that is, income generated through legacies (particularly significant), gifts, donations, grants, membership subscriptions and sponsorships. One-sixth (15%) of the total income was from activities for generating funds, such as jumble sales, lotteries and charity shops. Less than 1/10 was raised through 'charitable activities', such as trading goods and services.

Figure 3 shows how average levels of the total income and expenditure have varied over time. League of Friends' incomes have been declining since 1995, on average by approximately £901 a year. The decline was most pronounced in 2011 and 2012, showing some signs of recovery in 2013 and 2014. Fixed-effects regression analysis, entering a constant term and year as the only predictors, in order to compare income with that of a base category (income in 1995), indicates that these temporal changes were statistically significant. The average and median income that Leagues of Friends have received yearly was on the rise until the mid-2000s but since then has declined; the decline in income in general has been statistically significant (b=−901.29, SE 230.20), p<0.01; constant b=1 785 435, SE 461 467, p<0.001.

There was, however, little concern among our case studies regarding declining levels of income. Indeed, some had limited their fundraising activities, either through recognising a potential crowding out of other local charities or due to uncertainty over the future of the hospital. Some League of Friend's members, however, questioned how long they could sustain current income levels, often explaining this with reference to declining membership levels. These were also seen as harbingers of more general reduced community support.

Expenditure had also declined, but less sharply than income (figure 3). Since 1995 community hospital Leagues of Friends had, in most years, received more than they had spent. The exceptions from this trend are years 1997 and 2011 when an average Friends group spent more than it received. In 2011, their funding position was at its weakest (expressed in terms of the gap between income and expenditure) since 1995 (figure 3).

These financial resources are used to provide various amenities for community hospitals: patient comforts

(found across all case studies), staff development, equipment, buildings, staff time (found in a small number of cases). There was uncertainty about where the boundary between voluntary and statutory support should be drawn. Some cases adopted a clear demarcation avoiding anything they felt to be a statutory responsibility (eg, buildings, equipment and staffing), others did not. Concerns were expressed about the blurring of boundaries:

> 'We are supposed to help the hospital for extras. This was how it started off. It has got to be very much more mainstream now because of lack of funds' CH9, Volunteer

### Practical resources: providing services through voluntary organisations

Various national and local voluntary organisations also contributed to community hospitals, providing some direct form of service to patients getting into hospital, during their stay, or on discharge. They also provided support for families and carers of hospital patients, or more general support for the hospital through, for example, running fundraising events. Many of these services were free to the hospitals, some were paid for; some were delivered by paid staff employed by the voluntary organisations and others by volunteers. Some voluntary groups, or members of staff from within them, were colocated within the hospital, providing important mutual benefits.

The interaction of community hospitals with such voluntary organisations varied. The general perception was that more could be done to strengthen the links between institutions and local organisations. Two barriers identified were a lack of knowledge of local voluntary and community groups, and a lack of time for hospital staff to identify and build relationships.

### Intellectual resources: giving voice through information, consultation and coproduction

Leagues of Friends in particular, but also other community groups and individuals, played a role in supporting the hospital through representing the hospital in the community and/or vice versa. This happened at different levels.

At a basic level, an important role for the League of Friends was to provide a communication channel between the hospital and its community. The Leagues of Friends helped raise the profile of the hospital and provide feedback on services. Many League of Friends members were well networked within their local communities, facilitating this role.

Leagues of Friends had come, by design or default, to represent the 'voice of the community' in various consultation processes concerning the future of the hospital/services. This was not always a role they had anticipated, nor was it one which all had the capacity to fulfil. Considerable frustration was expressed about such consultations, focusing on: tokenism, poor processes, lack of

timely communication, uneven engagement and an apparent inability to influence outcomes. Commissioners recognised the importance of consultation, suggesting communities would react 'more reasonably' to change if they had been involved in decisions. Some commissioners, however, expressed their own frustrations about the consultation processes:

> 'however much you do it's never enough and some people will always feel that they've in some way been excluded.' (CH8&9, Commissioner)

Deeper, sustained community involvement in the ongoing coproduction of community hospitals services was limited. This led to frustration that opportunities were being missed to draw on the expertise and energy of the community:

> 'There is no local input at all and I think there should be. You call it a local reference group: I would love to have a small local reference group and that would at least—not run the hospital—but at least be able to have a say and it could be listened to' (CH2, Volunteer)

Among the nine case studies, however, there were two significant exceptions. In one, following a decision by the NHS to close the hospital, the local community established a charity and ran a successful fundraising campaign to reopen it. The charity now owns and runs the hospital, with services provided by the NHS, and has established a range of other health and social care services on the same site. In another, again following (perceived) threats to the future of the hospital, the community formed a local action group which has become a key player in a now well established local health forum which brings together statutory health and social care providers with the local community to codesign and codeliver services.

### Making a difference

Distilling the combined impact of these resources is difficult. Some suggested that community involvement was 'key to the community hospital': it was what makes a hospital a 'community hospital'; community hospitals would be a 'totally different place' without it. In one, however, it was suggested that if the community did not get involved no-one would really notice, although 'some of the trimmings might disappear round the edge'. The study identified five sets of outcomes:

▶ Enhancing community hospital utilisation and resilience: Community engagement, it was suggested, may be an important factor in the apparent resilience of community hospitals.[5] Community involvement had also contributed to service utilisation in some hospitals, through raising awareness of facilities available. However, a tendency towards resisting change was seen by some to be a downside of community engagement, potentially preventing improvements to service delivery.

- ► Contributing to patient experience: Enhanced hospital buildings and equipment, resulting from voluntary efforts, were agreed to contribute to patient experience, giving them the 'extra bits which the hospital won't or can't afford', and contributing to the 'happy atmosphere' found in most of the case studies. It was suggested that some patients found it easier to talk to volunteers ('someone out of uniform'*)* than to staff. Volunteers could help to tackle loneliness and boredom, and more generally to 'lift peoples' spirits' and 'making them happy'. Community engagement was also thought to be beneficial at the point of discharge, easing the transition home.

- ► Boosting staff morale: Staff told us that community involvement had an effect on them. The perception of being supported and valued by the community contributed to an identified distinction between the experience of working in a community hospital and an acute hospital, could help boost staff morale, and build loyalty to the hospital. On the other hand, some staff reported having to invest valuable time in facilitating community engagement, 'pick up' tasks when volunteers had not turned up or had not done a job effectively, and deal with the expectations of community members that arose from their support of the hospital (eg, an entitlement to access beds).

- ► Enhancing volunteer well-being: Engaging with the hospital also had outcomes for the volunteers themselves. Volunteering provided structure for some, and acted as a replacement for paid work, or as an alternative to it. The social interaction, physical and mental activity associated with volunteering was important for some, particularly older, respondents. However, for some, volunteering encroached on personal and family time and could be experienced as stressful and tiring, particularly when it proved difficult to recruit enough helpers.

- ► Community well-being: the activities involved in supporting community hospitals were building, and often reliant on social interaction, networks and trust. Fundraising events in particular were highlighted as important functions in the social calendar of some communities, bringing people together and tackling social isolation. This was seen as particularly important for older people.

## DISCUSSION

Community hospitals are generally well supported by their local communities. Many benefit from considerable input of resources in terms of time, money, service and intelligence. Together through this voluntary 'energy', communities provide significant additional resources to their community hospitals. As previous studies have suggested, these resources can positively affect patient experience[11 21 22] and service quality[14] by adding capacity, enhancing facilities and boosting staff morale.

However, levels of voluntary income have been declining. Clifford likewise found Leagues of Friends were one of the groups of charities that had experienced a decline in income.[23] Membership has also declined. And it was increasingly hard to recruit volunteers. While the study identified widespread latent support for community hospitals, regular, ongoing, active involvement was often limited to a relatively small group of volunteers. Getting new and particularly younger people to actively engage—beyond moments of crisis—was a challenge. Respondents in part attributed these challenges to changes within community hospital services which were felt to make them less accessible or relevant to the local—particularly younger—community,[5] and in part to wider societal changes which were generally felt to be making community engagement more difficult (although it is worth noting that long term national levels of volunteering are static[24]). The variance in levels of community support, between and within communities and over time, echoes studies of voluntary support in other areas of healthcare[14 25] as well as in other sectors.[26 27] These findings question the extent to which voluntary support can be always be considered a sustainable source of 'renewable energy'.

We agree with Munoz *et al*'s (p.221) conclusion that 'harnessing more local volunteers […] is more complex than governments assume'.[25] However, we found limited evidence of extensive efforts being made to do so. Several limits to the involvement of volunteers and voluntary groups were identified, alongside a general lack of 'investment' in their engagement. Converting the extensive (often passive) support found across communities into more regular, active engagement requires investment of time, enthusiasm and inevitably money. Just as harnessing other sources of renewable energy (eg, wind, sun) requires investment in the appropriate infrastructure, so too—we suggest—does voluntary and community support. There was evidence from some of our case studies that sustained efforts to encourage and nurture voluntary support could, however, be effective.

This study has provided original insights into voluntary support for community hospitals, demonstrating variations in its contribution, its reliance on a relatively small number of committed individuals, and raises questions as to the extent to which it can be considered sustainable or 'renewable'. This provides important lessons for the policies predicated on the expansion of voluntary and community engagement in healthcare. Communities can be a significant resource for healthcare services, in ways which can enhance patient experience and service quality, but this resource should not be taken for granted. The scale of voluntary support—volunteering, voluntary income or coproductive activities—is unlikely to grow without considerable effort at national and local level to support it.

Study limitations include limits to the secondary data available for the detailed analysis of levels of volunteering (CC records exclude trustees and those who volunteer within community hospitals outside of the League of Friends) and expenditure from charitable funds. The

study was also limited by its scope: while we were able to focus in detail on community hospitals, we were not able to fully consider the wider system within which they are embedded.

Future research priorities include longitudinal research to explore changes in the levels and forms of voluntary support and comparative studies of voluntary support in different healthcare settings.

**Acknowledgements** We would like to thank all our study respondents who contributed to this study through participating in interviews, focus groups, site visits and responding to requests for information. Particular thanks go to matrons/ward sisters, community hospital clinicians and league of friends members who coordinated research activities for us, organised local reference groups and participated in learning events. We would also like to acknowledge the inputs of other members of the study team who were involved in the wider study design and data collection but were not directly involved in authoring the article: Nick Le Meseurier, Helen Tucker, Jon Glasby, Iestyn Williams, David Seamark, Tessa Crilly, John Crilly and Jan Marriot. Thanks also go to our funders, the National Institute for Health Research.

**Contributors** All authors met the ICMJE criteria for authorship. All made substantial contributions to the design and analysis: AEP and DD led on the qualitative aspects of the study, DK and JM on the quantitative. All were involved in drafting and/or revising the article, with AEP taking the lead. All have approved the final version of the paper, and are accountable for all aspects of the work.

**Funding** This work was supported by The National Institute for Health Research—Health Services and Delivery Research: New Research on Community Hospitals, grant number 12/177/13.

**Competing interests** None declared.

**Patient consent for publication** Not required.

**Ethics approval** The Wales Research Ethics Committee 6 reviewed and approved this research study on the 24th February, 2016, reference: 16/WA/0021.

**Provenance and peer review** Not commissioned; externally peer reviewed.

**Data availability statement** Quantitative data are available in a public, open access repository.

**ORCID iD**
Angela Ellis Paine http://orcid.org/0000-0002-4385-5098

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
