## [Reviewer comments · BMJ Open]

ARTICLE DETAILS

TITLE (PROVISIONAL)	Communities as 'renewable energy' for health care services? A multi-methods study into the form, scale and role of voluntary support for community hospitals in England
AUTHORS	Ellis Paine, Angela; Kamerāde, Daiga; Mohan, John; Davidson, Deborah

VERSION 1 – REVIEW

REVIEWER	Jingheng CAI Sun Yat-sen University P.R.CHINA
REVIEW RETURNED	30-Mar-2019

GENERAL COMMENTS	The manuscript investigates the different ways in which communities are involved in and support their community hospitals. The objective of the study and the designs are well presented. However, I have a major concern on the regression analysis. Page 9. The authors employed fixed-effects regression analysis, entering a constant term and year as the only predictors, in order to compare income with that of a base category. Given that the incomes from different years are correlated, it is not appropriate to analysis such data with a regression. Is it a linear growth model?
--

REVIEWER	Kenneth Chambaere Vrije Universiteit Brussel (VUB) & Ghent University, Belgium
REVIEW RETURNED	11-Apr-2019

GENERAL COMMENTS	The authors and researchers have offered a credible account of volunteer support (i.c. in UK community hospitals) which is notoriously difficult to study given the lack of registration and monitoring. I applaud this research project and its promoters, and offer here my suggestions to further improve the quality and accessibility of the paper. INTRODUCTION - For non-UK readers it would be good to clarify a little more the different services, associations and initiatives: the Leagues of Friends, community hospitals (yes, even these, what makes them community hospitals instead of general hospitals or other hospital types? what is their coverage?), Five Year Forward View, HelpForce initiative, acute Trusts, Charity Commission. Perhaps a clarifying box is an option.- I'm missing a bit the explicit research question as well as the outright justification for this study: why is it important to do this
--

study, answer this research question? Filling up knowledge gaps is good, as long as that leads to potential for change. The argumentation is in there implicitly but better to formulate it. I am already convinced, but not everyone may be without this justification

METHODS

- there are 2 different methodologies employed: best to separate them by subtitles.
- the explanation of the qualitative research method seems limited: perhaps the authors could check the COREQ guidelines to ensure all relevant information has been included.
- 9 case studies: have the researchers included a check with some of the 287 non-included community hospitals whether the findings deviate significantly with their situation or experience? Or are these the Local Reference Groups? How many such groups were there?

RESULTS

- "These figures are likely to be underestimates: they exclude trustees and those who volunteer within community hospitals but outside the League of Friends; and the calculations used only national averages for volunteer hours and the minimum national hourly wages." >> so this is very much a rough estimate of volunteering time and wage equivalent. Needs to be mentioned in the limitations. Perhaps this assertion also needs to be removed from the results as it is already offering interpretation of findings (to be avoided in a results section)
- Dito for "Not all volunteering resources are renewable."
- the seven identified volunteer roles are not quite all immediately clear to me. Some more clarification as to the content of "service delivery; facilities; and patient well-being" and differences between them would be helpful.
- reading the qualitative results, it would seem to me there is a plethora of data and themes crammed into this paper. This means that per theme only the most general insights can be reported and only the surface is scratched for each. I do hope that separate detailed reports are planned to elaborate on these general findings. If so, the authors could consider mentioning this toward the end of the paper.
- any chance of showing the Fixed-effects regression analysis results in a table? It is important enough to devote a table to it.
- it is unclear to me whether the researchers shared the Leagues of Friends results with the nine case studies participants, or whether they spoke from their own experience or their own local numbers. As such it is difficult to judge certain assertions in the results, such as "some questioned the future sustainability of income levels, and raised concerns about declining membership levels, which was seen as a harbinger of more general reduced community support."
- "A lack of knowledge or capacity amongst hospital staff to identify and build relationships with relevant organisations was identified as a barrier.": if there is room, it would be interesting to hear what respondents meant by knowledge or capacity of staff.
- reviewing the bigger picture, the seven volunteer roles identified in the human resources subsection largely coincide with the four types of resources. So the question becomes whether it is still worthwhile to report the seven roles as a result under human resources. It could be seen as a form of duplication of the result.

	- "Enhancing community hospital utilisation and resilience": there is no mention of carers in the text itself. - "Boosting staff morale": did this come from staff members directly or from others? A bit dubious if the latter... DISCUSSION - I believe it would be worthwhile to explore or speculate about the reasons for the dwindling number of volunteer memberships and income a bit more. That will aid considerably in judging the question of whether they are an infinitely renewable resource. What are the prerequisites to a critical mass of volunteer support? There will be practical reasons such as a lack of focus on recruitment/fundraising (less threatening because easily overturned), as the authors posit. But, perhaps there are also societal reasons such as social fragmentation, lack of cohesion, intolerance, etc. (more threatening as fundamental to volunteering support). Are there indications for such a trend? - "This provides important lessons for policies predicated on the expansion of voluntary and community engagement in health care." > which are those lessons? What are the policy recommendations here? And recommendations for the community hospitals? - the limitations part is very limited
--	--

REVIEWER	Helen Roberts Faculty of Medicine, University of Southampton, UK
REVIEW RETURNED	15-Apr-2019

GENERAL COMMENTS	This is a well-written and interesting paper which addresses important questions around continuing voluntary support for community hospitals. As part of a large national multi-method study, data was gathered relating to 358 financial years for 245 community based charities. Nine qualitative case studies carried out representing a diverse range of community hospitals, comprised interviews and focus groups with a wide range of stakeholders. PPI involvement was high including study design. The results are presented thematically and highlight areas of challenge and success for community hospitals. I have a few comments:  1. The main limitation of the quantitative dataset is that only around 25% of the sample was included each year, perhaps because only those with an income >£25k were included. Thus smaller hospitals may not have been so well represented and it would be important for the authors to comment on the extent to which they have been included in the case studies. 2. The financial information was collected between 1995 and 2014, and a description of old much of the data was, would be useful eg mean (SD) or median (IQR) time lag to current date.
---

REVIEWER	Larkin Lamarche McMaster University, Canada
REVIEW RETURNED	16-Apr-2019

GENERAL COMMENTS	This study examined the various forms and levels of community and voluntary support for community hospitals in England. It is clear that the study examines a piece of the healthcare system that is typically understudied, thus contributing to the broader literature. I have several concerns to consider and/or address prior to a recommendation of publication.
---

	Background  1. The background gave good historical context to the study. A definition of 'renewable energy' in the sense of volunteers would be helpful; perhaps articulating what would it mean for this source to be renewable in terms of what you examined in the study. This may help to better clarify your research question and how it relates to the data collection methods used. 2. How is 'renewable' different from 'sustainable'? Sustainable is a very common word in implementation science and in the healthcare setting generally. Will using a word like renewable miss those who study sustainability?  a. It may be useful to situate the study in the vast literature on sustainability within healthcare. My guess is that there is still not much in terms of this particular setting or perspective, but it would be helpful to place it within the broader literature. b. This may also help in placing boundaries around the study – the sustainability, and perhaps the extent to which communities are a renewable energy source, likely depends on factors within the larger system. There are some factors listed here: Crisp Nigel. What would a sustainable health and care system look like? BMJ 2017; 358 :j3895 https://www.bmj.com/content/358/bmj.j3895 c. Perhaps a limitation to the study is that it does not consider the wider system? 3. A definition of 'volunteer' would help readers understand this role - there are many definitions of volunteers in the literature, and not all articles will use the term 'volunteer' (i.e., community worker, peer, peer worker, lay worker, peer system navigator, peer support, coach, community volunteer, etc). Was your definition considered in the inclusion/exclusion of articles? 4. Add 'scale' as part of the research objective in the abstract (to match title and speak to the 'renewable' energy). I feel 'scale' is essential in the research objective as this appears to be the 'mixed-methods' research aim. Methods  1. What did you base the deductive coding on (p. 6)? 2. References should be included in the data analysis section for your qualitative data – some leaders in analysis are Braun & Clarke – for thematic analysis: Braun, V., Clarke, V., Hayfield, N., & Terry, G. (2019). Thematic analysis. Handbook of Research Methods in Health Social Sciences, 843-860. 3. Please articulate ways rigor was fostered in the data analysis (trustworthiness, authenticity, and others that may apply). Please provide references for these methods. 4. Is there any more articulation that can be made in terms of 'mixing' in terms of design (convergent, explanatory sequential, etc.)? I appreciate the mixing in terms of data collection, however, it does appear that conclusions were based on convergence from analysis. This may mean having a research aim for the quantitative data, another for the qualitative data and one for the mixed data.
--	---

	Having a rationale section for using mixed methods may also be important (and is essential for this type of research). Some good sources for mixed-methods: Creswell is a well known name in this area, Creswell, J. W., & Clark, V. L. P. (2017). Designing and conducting mixed methods research. Sage publications; Plano Clark, V. L., & Ivankova, N. V. (2016). Mixed methods research: A guide to the field. Los Angeles, CA: Sage. a. Good reporting of a mixed methods study (GRAMMS) could be used, or similar: O'cathain, A., Murphy, E., & Nicholl, J. (2008). The quality of mixed methods studies in health services research. Journal of Health Services Research & Policy, 13(2), 92-98. http://brown.uk.com/teaching/HEST5001/ocathain.pdf 5. The interview/focus group question guide should be included as an appendix or table. If this is part of a larger study with other aims, the questions where data from the present study came from should be included at minimum. 6. Does any positionality need to be made in terms of the researchers link to the community hospitals or community hospital sector? I see connection between the research team and the community hospital as beneficial in terms of analysis, however, should be explicitly stated to add context. Results 1. Were there any instances of divergence? Discussion 1. See point about limitation under 'Background' (2c above). In fact, I would delete the one limitation about your interviewees and connection to the community hospital. I see this as a strength in terms of purposive sampling – more depth can be gained from talking to people involved in the community hospital, and the findings suggest that these individuals can speak to what's working and what's not working. Minor points 1. Add sustainability as a key word 2. I think 'lestyn Williams' needs a capital 'L'
--	---

VERSION 1 – AUTHOR RESPONSE

Reviewer: 1

Page 9. The authors employed fixed-effects regression analysis, entering a constant term and year as the only predictors, in order to compare income with that of a base category. Given that the incomes from different years are correlated, it is not appropriate to analysis such data with a regression. Is it a linear growth model?

Response: Thank you very much to the reviewer for highlighting that we have not been explicit enough when describing the regression methods used in this paper. This study used fixed effects (FE) regression model to examine how within-hospital charity changes in time (years) are linked to changes in the annual income within each individual charity over time, while eliminating unobserved heterogeneity – confounding effects from time-constant variables (e.g. Halaby, C. H. (2004). Panel models in sociological research. *Annual Review of Sociology*, 30, 507-544). The year variable examines any variation in the outcome that happen over time. We have now included this more detailed description of the analysis methods used in the Methods section (P6)

Reviewer: 2

- For non-UK readers it would be good to clarify a little more the different services, associations and initiatives: the Leagues of Friends, community hospitals (yes, even these, what makes them community hospitals instead of general hospitals or other hospital types? what is their coverage?), Five Year Forward View, HelpForce initiative, acute Trusts, Charity Commission. Perhaps a clarifying box is an option.

Response: Thank you for highlighting this. We have modified the text, adding further explanations or deleting unnecessary terms. (P4, and elsewhere as relevant)

- I'm missing a bit the explicit research question as well as the outright justification for this study: why is it important to do this study, answer this research question? Filling up knowledge gaps is good, as long as that leads to potential for change. The argumentation is in there implicitly but better to formulate it. I am already convinced, but not everyone may be without this justification

Response: We have more clearly stated the question, and the reason why it is important in the closing paragraph of the introduction and the opening of the methods section. (P5)

METHODS

- there are 2 different methodologies employed: best to separate them by subtitles.

Response: Rather than adding subtitles, we have more clearly indicated which paragraphs deal with the quantitative methods and which with the qualitative. (P5&6)

- the explanation of the qualitative research method seems limited: perhaps the authors could check the COREQ guidelines to ensure all relevant information has been included.

- Response: Thank you. We have added in additional details, including reference to the SRQR guidelines which we followed. We have also reinforced the reference to the report of the full study which provides much greater detail of all the methods. (P5&6)

- 9 case studies: have the researchers included a check with some of the 287 non-included community hospitals whether the findings deviate significantly with their situation or experience? Or are these the Local Reference Groups? How many such groups were there?

-Response: There were nine local reference groups – one for each of the case studies. Beyond checking the emerging findings with each of the nine community hospitals (through the local reference groups), we also shared and discussed emerging findings with the Community Hospital Association committee and members, allowing us to check the validity of our findings with non-case study hospitals. We have added reference to this within the article (p6-8)

RESULTS

- "These figures are likely to be underestimates: they exclude trustees and those who volunteer within community hospitals but outside the League of Friends; and the calculations used only national averages for volunteer hours and the minimum national hourly wages." >> so this is very much a rough estimate of volunteering time and wage equivalent. Needs to be mentioned in the limitations. Perhaps this assertion also needs to be removed from the results as it is already offering interpretation of findings (to be avoided in a results section)

Response: thank you. We have removed from the findings sections, replacing it with a findings from the case studies which highlighted additional volunteers within the community hospitals outside of the League of Friends and therefore not included within these figures. (P8 &16)

- Dito for "Not all volunteering resources are renewable."

Response: Removed

- the seven identified volunteer roles are not quite all immediately clear to me. Some more clarification as to the content of "service delivery; facilities; and patient well-being" and differences between them would be helpful.

Response: see below ('the bigger picture' comment)– we have provided more explicit examples of roles undertaken by volunteers.

- reading the qualitative results, it would seem to me there is a plethora of data and themes crammed into this paper. This means that per theme only the most general insights can be reported and only the surface is scratched for each. I do hope that separate detailed reports are planned to elaborate on these general findings. If so, the authors could consider mentioning this toward the end of the paper.

Response: Many thanks for your interest. This was a very rich study with lots of findings. We have pointed readers towards the main study report, which does elaborate on some of the general findings discussed here. We have begun to write additional articles addressing separate, but related questions, which would expand upon some of the themes touched upon within this article, but as references for those are not yet available we did not feel it appropriate to mention them.

- any chance of showing the Fixed-effects regression analysis results in a table? It is important enough to devote a table to it.

Response: Thank you very much for the recommendation to show the regression analysis results in a table. As the fixed-effects regression model included only the constant and the year (as an interval variable), it does not warrant the space of using a table to present these results and therefore to address your comment, we have instead added other parameters that would normally be presented in a table in the text: "the decline in income in general has been statistically significant ($b = -901.29$, $SE = 230.20$), $p < .01$; constant $b = 1785435$, $SE = 461467$, $p < .001$)" (P10)

- it is unclear to me whether the researchers shared the Leagues of Friends results with the nine case studies participants, or whether they spoke from their own experience or their own local numbers. As such it is difficult to judge certain assertions in the results, such as "some questioned the future sustainability of income levels, and raised concerns about declining membership levels, which was seen as a harbinger of more general reduced community support."

Response: Emerging findings were shared with case study participants, through the local reference groups and in conversation, but not explicitly within the interviews. These statements were based on the League of Friends' own knowledge of their fluctuating income and membership levels. We have clarified this within the text. (P11)

- "A lack of knowledge or capacity amongst hospital staff to identify and build relationships with relevant organisations was identified as a barrier.": if there is room, it would be interesting to hear what respondents meant by knowledge or capacity of staff.

Response: We are limited by word counts, but have added a few words to clarify this P12. It was relating to a lack of knowledge of what community and voluntary groups operated in the local area and how they might be (usefully) involved in supporting the hospital, and/or a lack of time to find this out, make contact and build relationships.

- reviewing the bigger picture, the seven volunteer roles identified in the human resources subsection largely coincide with the four types of resources. So the question becomes whether it is still

worthwhile to report the seven roles as a result under human resources. It could be seen as a form of duplication of the result.

Response: Taking this and the above comment re volunteer roles together, we have edited the text to include a few specific examples of the kinds of roles undertaken by volunteers. (P9)

- "Enhancing community hospital utilisation and resilience": there is no mention of carers in the text itself.

Response: We have removed reference to 'carers' within the heading. Although there was evidence of the impact of community support on carers, we did not have space to expand upon or give explicit examples of that within the article, so we have removed reference to carers and kept the focus on patients.

- "Boosting staff morale": did this come from staff members directly or from others? A bit dubious if the latter...

Response: this came directly from staff (although was also suggested by others). We have made this clearer within the text. (P14)

DISCUSSION

- I believe it would be worthwhile to explore or speculate about the reasons for the dwindling number of volunteer memberships and income a bit more. That will aid considerably in judging the question of whether they are an infinitely renewable resource. What are the prerequisites to a critical mass of volunteer support? There will be practical reasons such as a lack of focus on recruitment/fundraising (less threatening because easily overturned), as the authors posit. But, perhaps there are also societal reasons such as social fragmentation, lack of cohesion, intolerance, etc. (more threatening as fundamental to volunteering support). Are there indications for such a trend?

Response: We have added some additional reflections on this into the discussion section. (P15-16)

- "This provides important lessons for policies predicated on the expansion of voluntary and community engagement in health care." > which are those lessons? What are the policy recommendations here? And recommendations for the community hospitals?

Response: We have expanded upon this. (P16)

- the limitations part is very limited

Response: We have developed these as per this and other reviewer comments. (P16)

Reviewer: 3

1. The main limitation of the quantitative dataset is that only around 25% of the sample was included each year, perhaps because only those with an income >£25k were included. Thus smaller hospitals may not have been so well represented and it would be important for the authors to comment on the extent to which they have been included in the case studies.

Response: The quantitative dataset did include LoF with lower incomes, but less detailed records are kept for smaller charities (levels of income and expenditure was available for all, but more detailed breakdowns of both were only available for those over £25k). Case studies were selected to include those with diverse voluntary income levels – ranging from one which had an average annual income over a five year period of £1,371 to one with an average of £423,521. We have noted this within the text. (P6)

2. The financial information was collected between 1995 and 2014, and a description of old much of the data was, would be useful eg mean (SD) or median (IQR) time lag to current date.

Response: Thank you very much to the reviewer for pointing out the differences in the age of data from different years. In our paper all financial figures for 1995-2014 were adjusted for inflation using the Office for National Statistics Retail Prices annual index; therefore all financial information in this paper is presented 'in 2014 prices'. We have added this explanation to the paper (p6)

Reviewer: 4

1. The background gave good historical context to the study. A definition of 'renewable energy' in the sense of volunteers would be helpful; perhaps articulating what would it mean for this source to be renewable in terms of what you examined in the study. This may help to better clarify your research question and how it relates to the data collection methods used.

Response: We have provided further clarification within the introduction. (P4-5)

2. How is 'renewable' different from 'sustainable'? Sustainable is a very common word in implementation science and in the healthcare setting generally. Will using a word like renewable miss those who study sustainability?

Response: This is very helpful, thank you. We had focused on 'renewable' as this was suggestion within the Five Year Forward View, but agree with the reviewer that 'sustainable' would have greater resonance. This is reflected within the revised paper. (p4-5)

a. It may be useful to situate the study in the vast literature on sustainability within healthcare. My guess is that there is still not much in terms of this particular setting or perspective, but it would be helpful to place it within the broader literature.

Response: Many thanks for this helpful suggestion. Unfortunately we do not have space within the article to fully engage with the wider literature on sustainability, but we have drawn on and now added into the article a reference to Nigel Crisp's thinking on sustainability (P4).

b. This may also help in placing boundaries around the study – the sustainability, and perhaps the extent to which communities are a renewable energy source, likely depends on factors within the larger system. There are some factors listed here: Crisp Nigel. What would a sustainable health and care system look like? BMJ 2017; 358 :j3895 <https://www.bmj.com/content/358/bmj.j3895>

Response: Thank you- as above.

c. Perhaps a limitation to the study is that it does not consider the wider system?

Response: This is a fair point: the study did not fully consider the wider system. We have added this as a limitation.

3. A definition of 'volunteer' would help readers understand this role - there are many definitions of volunteers in the literature, and not all articles will use the term 'volunteer' (i.e., community worker, peer, peer worker, lay worker, peer system navigator, peer support, coach, community volunteer, etc). Was your definition considered in the inclusion/exclusion of articles?

Response: Thank you. We have been guided by the United Nations broad definition of volunteering as encompassing "a wide range of activities, undertaken of free will, for the general public good and where monetary reward is not the principal motivating factor." Source: UN Volunteers (2015) State of the world's volunteerism report. United Nations Volunteers. As the reviewer notes, many activities which would be counted as volunteering within this definition are not labelled as such within publications and not always thought of as such by participants. Given the different elements of voluntary support that are included in the paper (volunteering, voluntary income/fundraising, co-production etc) we did not have space to provided definitions of each.

4. Add 'scale' as part of the research objective in the abstract (to match title and speak to the 'renewable' energy). I feel 'scale' is essential in the research objective as this appears to be the 'mixed-methods' research aim.

Response: Thank you. We have made this explicit within the abstract and elsewhere.

Methods

1. What did you base the deductive coding on (p. 6)?

Response: We have removed reference to deductive coding. During early stages of analysis we worked inductively to develop the codes and themes, but as analysis progressed we worked deductively to look at our data in light of those codes and themes to review whether additional evidence was needed or could be found within the interviews to support them. We have provided additional details of the coding process. (P7)

2. References should be included in the data analysis section for your qualitative data – some leaders in analysis are Braun & Clarke – for thematic analysis: Braun, V., Clarke, V., Hayfield, N., &

Terry, G. (2019). Thematic analysis. *Handbook of Research Methods in Health Social Sciences*, 843-860.

Response: Thank you. We had based our thematic analysis on Braun and Clarke's six steps, but had not made this explicit within the text. This has now been done. P7

3. Please articulate ways rigor was fostered in the data analysis (trustworthiness, authenticity, and others that may apply). Please provide references for these methods.

Response: Additional details of the analysis process, reference to our use of SRQR have been added (P5-7)

4. Is there any more articulation that can be made in terms of 'mixing' in terms of design (convergent, explanatory sequential, etc.)? I appreciate the mixing in terms of data collection, however, it does appear that conclusions were based on convergence from analysis. This may mean having a research aim for the quantitative data, another for the qualitative data and one for the mixed data. Having a rationale section for using mixed methods may also be important (and is essential for this type of research). Some good sources for mixed-methods: Creswell is a well known name in this area, Creswell, J. W., & Clark, V. L. P. (2017). *Designing and conducting mixed methods research*. Sage publications; Plano Clark, V. L., & Ivankova, N. V. (2016). *Mixed methods research: A guide to the field*. Los Angeles, CA: Sage.

Response: Thank you. We had added additional details, and included references to a couple of the resources which we used to guide our work in this area (P5-7)

a. Good reporting of a mixed methods study (GRAMMS) could be used, or similar: O'cathain, A., Murphy, E., & Nicholl, J. (2008). The quality of mixed methods studies in health services research. *Journal of Health Services Research & Policy*, 13(2), 92-98.

<http://brown.uk.com/teaching/HEST5001/ocathain.pdf>

Response: Thank you. We found O'cathain a useful resource, and have now included reference to this article. (p5)

5. The interview/focus group question guide should be included as an appendix or table. If this is part of a larger study with other aims, the questions where data from the present study came from should be included at minimum.

Response: Topic guides have been added as a supplementary file as per editors instructions.

6. Does any positionality need to be made in terms of the researchers link to the community hospitals or community hospital sector? I see connection between the research team and the community hospital as beneficial in terms of analysis, however, should be explicitly stated to add context.

Response: We have added a sentence about the different positionalities of the qualitative research team in relation to their experience of health care systems and voluntary sector/volunteering.

Results

1. Were there any instances of divergence?

Response: We have noted instances of divergence, where they occurred – such as where one case study had reversed a downward trend in its engagement with the League of Friends

Discussion

1. See point about limitation under 'Background' (2c above). In fact, I would delete the one limitation about your interviewees and connection to the community hospital. I see this as a strength in terms of purposive sampling – more depth can be gained from talking to people involved in the community hospital, and the findings suggest that these individuals can speak to what's working and what's not working.

Response: Thank you. We agree. This was a limitation of the wider study, when we also considered questions of the value of the community hospital to the local community, but is relevant to the specific question being addressed by this paper. We have removed it as a limitation.

Minor points

1. Add sustainability as a key word

Response: Added – thank you

2. I think 'lestyn Williams' needs a capital 'L'

Response: lestyn, is capital 'l' rather than 'L'

VERSION 2 – REVIEW

REVIEWER	Jingheng CAI Sun Yat-sen University P.R.CHINA
REVIEW RETURNED	27-Jul-2019

GENERAL COMMENTS	I have no further comments.
-----------------------------

REVIEWER	Kenneth Chambaere End-of-Life Care Research Group, Vrije Universiteit Brussel (VUB) & Ghent University, Belgium
REVIEW RETURNED	15-Jul-2019

GENERAL COMMENTS	The authors have adequately addressed my suggestions. This interesting paper has improved and definitely merits publication.
--

REVIEWER	Larkin Lamarche McMaster University, Canada
REVIEW RETURNED	12-Aug-2019

GENERAL COMMENTS	I have no further comments to the authors. I believe they have addressed my concerns fully. Findings will contribute to the literature in a unique way.
---